# Phylogenomic Reconstruction and Metabolic Potential of the Genus *Aminobacter*

**DOI:** 10.3390/microorganisms9061332

**Published:** 2021-06-19

**Authors:** Irene Artuso, Paolo Turrini, Mattia Pirolo, Gabriele Andrea Lugli, Marco Ventura, Paolo Visca

**Affiliations:** 1Department of Science, Roma Tre University, Viale G. Marconi 446, 00146 Rome, Italy; irene.artuso@uniroma3.it (I.A.); paolo.turrini@uniroma3.it (P.T.); 2Department of Veterinary and Animal Sciences, University of Copenhagen, Stigbøjlen 4, 1870 Frederiksberg C, Denmark; mapi@sund.ku.dk; 3Laboratory of Probiogenomics, Department of Chemistry, Life Sciences, and Environmental Sustainability, University of Parma, Parco Area delle Scienze 11a, 43124 Parma, Italy; gabrieleandrea.lugli@unipr.it (G.A.L.); marco.ventura@unipr.it (M.V.); 4Interdepartmental Research Centre “Microbiome Research Hub”, University of Parma, 43124 Parma, Italy

**Keywords:** *Aminobacter anthyllidis*, glyphosate, methylamine degradation, methylotrophy, methyl halide, nitrogen fixation, nodulation, phosphonates, whole genome sequencing

## Abstract

Bacteria belonging to the genus *Aminobacter* are metabolically versatile organisms thriving in both natural and anthropized terrestrial environments. To date, the taxonomy of this genus is poorly defined due to the unavailability of the genomic sequence of *A. anthyllidis* LMG 26462^T^ and the presence of unclassified *Aminobacter* strains. Here, we determined the genome sequence of *A. anthyllidis* LMG 26462^T^ and performed phylogenomic, average nucleotide identity and digital DNA-DNA hybridization analyses of 17 members of genus *Aminobacter*. Our results indicate that 16S rRNA-based phylogeny does not provide sufficient species-level discrimination, since most of the unclassified *Aminobacter* strains belong to valid *Aminobacter* species or are putative new species. Since some members of the genus *Aminobacter* can utilize certain C1 compounds, such as methylamines and methyl halides, a comparative genomic analysis was performed to characterize the genetic basis of some degradative/assimilative pathways in the whole genus. Our findings suggest that all *Aminobacter* species are heterotrophic methylotrophs able to generate the methylene tetrahydrofolate intermediate through multiple oxidative pathways of C1 compounds and convey it in the serine cycle. Moreover, all *Aminobacter* species carry genes implicated in the degradation of phosphonates via the C-P lyase pathway, whereas only *A. anthyllidis* LMG 26462^T^ contains a symbiosis island implicated in nodulation and nitrogen fixation.

## 1. Introduction

Members of the genus *Aminobacter* (family *Alphaproteobacteria*) are soil bacteria described as motile aerobic rods with a strictly respiratory metabolism, capable of growing heterotrophically in the presence of a wide variety of organic substrates and colonizing natural and anthropized terrestrial ecosystems [1].

The taxonomy of the genus *Aminobacter* has a recent history of debate and change [2,3,4]. The first known member of the genus was a methylamine-utilizing bacterium isolated from soils enriched with various amines [5]. This bacterium was originally assigned to the genus *Pseudomonas* and named *Pseudomonas aminovorans* for its ability to utilize various amines as sole carbon and energy sources [5]. In 1990, tetramethylammonium hydroxide- and *N*,*N*-dimethylformamide-utilizing bacteria (strains TH-3^T^ and DM-81^T^, respectively) were isolated from soils contaminated with industrial solvents [6,7] and found to resemble *P. aminovorans* [6,7]. Following chemotaxonomic investigations, Urakami et al. [2] proposed the transfer of these bacteria to a new genus within the *Phyllobacteriaceae* family and coined the name *Aminobacter* to denote the genus ability to catabolize methylated amines [1,5]. The genus *Aminobacter* was initially composed of *Aminobacter aminovorans* (basonym *P. aminovorans* [2]) as type species, together with *Aminobacter aganoensis* (strain TH-3^T^ = DSM 7051^T^) and *Aminobacter niigataensis* (strain DM-81^T^ = DSM 7050^T^) as novel species [2]. Subsequently, two strains, that is, IMB-1^T^ and CC495^T^, isolated from a fumigated agricultural soil and a beech forest soil, respectively [8,9], were included in genus *Aminobacter* with the species name *Aminobacter ciceronei* (strain IMB-1^T^ = DSM 15910^T^ = DSM 17455^T^) and *Aminobacter lissarensis* (strain CC495^T^ = DSM 17454^T^) [10], both able to utilize methyl halides as sole carbon and energy sources. In 2012, the genus *Aminobacter* was further expanded by the addition of the new monotypic species, *Aminobacter anthyllidis*, whose type strain LMG 26462^T^ originated from the root nodule of the plant *Anthyllis vulneraria* [11]. Recently, the genus *Carbophilus* was incorporated into the genus *Aminobacter* [3], and *Aminobacter carboxidus* DSM 1086^T^ (basonym *Carbophilus carboxidus* [12]) was subsequently classified as a distinct strain of the species *A. lissarensis* [4].

From a metabolic perspective, *Aminobacter* species are a new group of facultatively methylotrophic bacteria due to their ability to utilize reduced carbon substrates containing no carbon-carbon bonds (C1 compounds) as sole carbon and energy source [13,14]. However, genetic information about the degradative and/or assimilatory pathways of C1 compounds in the genus is so far limited.

*A. aminovorans*, *A. aganoensis* and *A. niigataensis* are known to degrade C1 methylated amines [1]. These reduced nitrogen compounds ([CH_3_]_n_NH_x_) are produced in nature as by-products of protein decomposition [15], and their oxidation takes place via two major routes: (i) direct oxidation to formaldehyde and ammonia by methylamine dehydrogenase (MADH), (ii) conversion of methylamine to N-methylglutamate (NMG), followed by the oxidation of NMG to methylene tetrahydrofolate (CH_2_=THF) [16,17]. The assimilation of C1 compounds can take place at the level of: (i) formaldehyde, via the ribulose monophosphate (RuMP) cycle; (ii) CO_2_, via the Calvin cycle or (iii) through the combination of CH_2_=THF and CO_2_, via the serine cycle [14].

*A. ciceronei* and *A. lissarensis* are both endowed with the ability to degrade methyl halides [8,9], a group of atmospheric pollutants that contribute to the destruction of the stratospheric ozone layer [18]. Methyl halide degradation occurs by the transfer of the methyl group to tetrahydrofolate, followed by oxidation to methylene tetrahydrofolate, thus generating reducing equivalents for biosynthesis [19]. Of note, methyltransferase gene clusters have been described in both *A. ciceronei* and *A. lissarensis* [20,21].

Genes encoding the form I carbon monoxide (CO) dehydrogenase (CODH), a heterotrimeric enzyme responsible for the oxidation of CO to carbon dioxide (CO_2_), were detected in *A. lissarensis* [4,22]. Indeed, *A. lissarensis* DSM 1086 is described as a “carboxydobacterium” for its ability to grow aerobically on CO as the sole carbon and energy source [12,23].

Recently, glyphosate degradation has been documented for *A. aminovorans* strain KCTC 2477 [24]. Glyphosate is a phosphonate herbicide containing a direct covalent bond between adjacent carbon and phosphorus atoms (the C-P bond) [25]. Many soil bacteria have developed sophisticated enzymatic machinery, known as carbon-phosphorus (C-P) lyase, which allows them to extract phosphate from a wide range of phosphonate compounds [26].

The genus *Aminobacter* also includes some unclassified strains with interesting bioremediation properties [27], namely *Aminobacter* sp. SR38 and *Amininobacter* sp. MSH1, capable of degrading the atrazine herbicide [28,29] and the 2,6-dichlorobenzamide (BAM) pesticide [30], respectively.

*A. anthyllidis* LMG 26462^T^ is somehow unique in the genus since it is the only species capable of fixing nitrogen from the atmosphere [11]. Preliminary nodulation and nitrogen fixation tests suggest that this monotypic species could represent the first, maybe the only, legume symbiont in the genus *Aminobacter* [11].

At present, the phylogeny of the genus *Aminobacter* is still uncertain due to the unavailability of the genomic sequence of *A. anthyllidis* LMG 26462^T^ and the presence of several strains presumptively assigned to the genus *Aminobacter*. In addition, while a broad metabolic potential for several members of *Aminobacter* genus has been suggested, the genetic basis of the catabolic/assimilatory pathways used by *Aminobacter* species to degrade recalcitrant compounds remains largely unknown. Two main questions can therefore be raised: (i) which are the evolutionary relationships among members of the genus *Aminobacter* and (ii) to which extent are the catabolic/assimilatory pathways conserved within the genus? Here, we report the draft genome sequence of *A. anthyllidis* LMG 26462^T^ together with a complete phylogenomic reconstruction of the whole genus *Aminobacter* with the aim to better delineate the taxonomic relationships between all *Aminobacter* species, including so far unclassified strains. Moreover, the gene clusters implicated in the catabolism of methylated amines, methyl halides and CO were investigated in all *Aminobacter* species, along with the degradation pathways of xenobiotic compounds, including atrazine, BAM and glyphosate. Assimilatory processes of monocarbon units and nitrogen fixation were also considered, to get a complete picture of the metabolic potential of the genus *Aminobacter*.

## 2. Materials and Methods

### 2.1. DNA Extraction and Genome Sequencing

*A. anthyllidis* LMG 26462^T^ was obtained from BCCM/LMG Bacteria Collection (Laboratorium voor Microbiologie, Universiteit Gent, Ghent, Belgium) and aerobically grown at 27 (±1) °C in Trypticase soy broth. DNA extraction was performed using a QIAamp DNA minikit (Qiagen). The quantity and quality of the extracted DNA were tested using a Thermo Scientific™ NanoDrop 2000 spectrophotometer (NanoDrop Technologies, Thermo Scientific) and by agarose gel electrophoresis, respectively. Additionally, the genomic DNA quality was evaluated by using Agilent TapeStation Systems 2200 (Agilent Technologies, Santa Clara, CA, USA). A genomic library of *A. anthyllidis* LMG 26462^T^ was obtained with the TruSeq DNA PCR-free sample preparation kit (Illumina, Inc., San Diego, CA, USA). Genome sequencing was performed with a NextSeq 500 sequencing system, according to the supplier’s protocol (Illumina, UK), and library samples were loaded into a midoutput kit v2.5 (300 cycles) (Illumina, UK), producing 1,642,495 paired-end reads. The raw sequence reads were filtered and trimmed using the command-line fastq-mcf software (https://expressionanalysis.github.io/ea-utils/ accessed on 1 March 2021). Fastq files of Illumina paired-end reads (150 bp) were used as input in the MEGAnnotator pipeline for microbial genome assembly and annotation [31]. This pipeline employed the program SPAdes v3.14.0 for de novo assembly of the genome sequence with the option “–careful” and a list of k-mer sizes of 21, 33, 55, 77, 99, 127 [32]. The genome quality was evaluated with the program CheckM v1.0.18 [33], estimating a genome completeness of 99.3% and 2.4% contamination. The contigs were submitted to the National Center for Biotechnology Information (NCBI) for the prediction of protein-encoding open reading frames (ORFs) and tRNA and rRNA genes using the NCBI Prokaryotic Genome Annotation Pipeline (PGAP) v5.2 [34]. The presence of genomic islands (GIs) was predicted by IslandViewer 4 [35], which uses SIGI-HMM, IslandPath-DIMOB and IslandPick prediction algorithms to generate a dataset of GIs.

### 2.2. Dataset Collection

Available genome sequences of Aminobacter strains were searched in the NCBI database, and 22 genome assemblies were retrieved, including type strains and unclassified Aminobacter spp. Redundant genomes were trimmed from the dataset, and only the last released genome for each strain was selected. The resulting dataset contained the genomes of 17 Aminobacter strains, including A. anthyllidis LMG 26462^T^ (Table 1). Plasmids were predicted from short-read data analysis with PlasmidSPAdes v3.14.0 program [36].

### 2.3. Phylogenetic Analysis of the 16S rRNA Gene

The 16S rRNA gene sequences of type strains for all validly published species belonging to the *Phyllobacteriaceae* family were retrieved from the NCBI database, and 16S rRNA gene sequences of unclassified *Aminobacter* strains were extracted from their genomes, when available. Alignment and phylogenetic analysis of the 16S rRNA gene sequences (Appendix A) were performed using MAFFT v7.48 [43], and positions containing gaps (585 positions) were removed using Gblocks v.0.91b [44], resulting in partial 16S rRNA gene sequences of 1098 nt in the final dataset. Due to the incomplete 16S rRNA gene sequence (919 bp), *Aminobacter* sp. J15 was excluded from the analysis. *Brucella melitensis* 16M^T^ was included as an outgroup in the analysis (Appendix A). Genetic distances were corrected using the Kimura two-parameter model [45], and a phylogenetic tree was constructed using the Neighbor-Joining (NJ) method [46], which was visualized using iTOL v6.1.2 [47]. The robustness of the phylogenetic tree was statically tested with a bootstrap of 1000 replicates [48].

### 2.4. Whole Genome-Based Phylogeny

A whole genome-based phylogeny was inferred for all available genome sequences of *Aminobacter* strains (n = 17) (Table 1), using the Type (Strain) Genome Server (TYGS) web-based pipeline (https://tygs.dsmz.de, accessed on 27 May 2021) [49]. *Mesorhizobium loti* DSM 2626^T^ (GenBank accession no. GCF_003148495.1) was included as an outgroup in the analysis. All pairwise comparisons among the set of genomes were conducted using Genome BLAST Distance Phylogeny (GBDP), and accurate intergenomic distances inferred under the algorithm “trimming” and distance formula d_5_ [50]; 100 distance replicates were calculated each. The resulting intergenomic distances were used to infer a balanced minimum evolution tree with branch support via FASTME v2.1.4 including SPR post-processing [51]. Branch support was inferred from 100 pseudo-bootstrap replicates each [48]. The tree was rooted in the selected outgroup and visualized using iTOL v6.1.2 [47].

### 2.5. Digital DNA–DNA Hybridization (dDDH) and Average Nucleotide Identity (ANI)

Species boundaries between *Aminobacter* members were investigated by the digital dDDH tool, as implemented in the Genome-To-Genome Distance Calculator (GGDC) v2.1 [50] and by whole genome ANI, as determined by FastANI v1.33 [52]. Average nucleotide identity and digital DNA–DNA hybridization were expressed as ANI and GGDC values, respectively. GGDC values > 70% in combination with ANI values > 96% were used as the boundary for species demarcation [53,54,55].

### 2.6. Detection and Structural Analysis of Nodulation and Nitrogen-Fixation Genes

The translated nucleotide sequences of *nif-fix* and *nod* genes from *Mesorhizobium japonicum* MAFF 303099^T^ (GCF_000009625.1) were individually used as queries in BLASTp v2.11.0 searches against the translated genome sequences of *A. anthyllidis* LMG 26462^T^ and *M. Japonicum* R7A (GCF_012913625.1). The presence of *nif-fix* and *nod* gene clusters was also investigated in all *Aminobacter* strains using the translated nucleotide sequences from *A. anthyllidis* LMG 26462^T^. Protein homologs were selected showing E-value < 10^−4^ and >60% identity across at least 80% of the protein sequence.

### 2.7. Detection and Structural Analysis of Methylotrophy Genes

The translated nucleotide sequences of the methylamine oxidizing genes and serine cycle genes from *Paracoccus aminovorans* JCM 7685^T^ (GCF_900005615.1) were individually used as queries in BLASTp v2.11.0 searches against the translated genome sequences of *A. aminovorans* DSM 7048^T^. The presence of the methylotrophy genes was investigated in all *Aminobacter* strains using the translated nucleotide sequences of the methylamine oxidizing genes and serine cycle genes from *A. aminovorans* DSM 7048^T^, the methyl halides oxidizing genes from *A. lissarensis* DSM 17454^T^ and *A. ciceronei* DSM 15910^T^, and the CODH and RuBisCO genes from *A. lissarensis* DSM 1086. Protein homologs were selected showing E-value < 10^−4^ and >60% identity across at least 80% of the protein sequence. When complete homologs were not detectable in draft genome sequences, partial coding sequences located at the end of the contigs were also considered.

### 2.8. Detection and Structural Analysis of Genes Implicated in Xenobiotic Degradation

The translated nucleotide sequences of genes implicated in glyphosate oxidation from *A. aminovorans* KCTC 2477, the atrazine degradation from *Aminobacter* sp. SR38, and the BAM degradation from *Aminobacter* sp. MSH1 were individually used as queries in BLASTp v2.11.0 searches against the translated genome sequences of all *Aminobacter* strains. Protein homologs were selected showing E-value < 10^−4^ and >60% identity across at least 80% of the protein sequence.

## 3. Results

### 3.1. Aminobacter Anthyllidis LMG 26462^T^ Genome Sequence

The draft genome sequence of A. anthyllidis LMG 26462^T^ consists of 6,717,907 bp, which results fragmented into 30 contigs with an N_50_ value of 670,596 bp, an average coverage of 113× and a mean GC content of 62.58%. Genome annotation identified 6486 ORFs, 51 tRNA genes and three rRNA genes.

### 3.2. 16S rRNA Gene-Based Phylogeny of Phyllobacteriaceae

A phylogenetic tree of 16S rRNA gene sequences (1098 positions) of 118 strains including all type strains from the *Phyllobacteriaceae* family and the selected *Aminobacter* strains, was obtained using NJ. *Aminobacter* sp. J15 was excluded from the analysis due to its incomplete 16S rRNA gene sequence (919 bp), which would lower the alignment length (aligned positions would decrease from 1698 to 666 bp). The phylogenetic analysis showed that most of the members of the *Aminobacter* genus form a discrete, statistically supported clade, clearly distinct from all other members of *Phyllobacteriaceae* (Figure 1). Notably, *Aminobacter* sp. J41 and *Aminobacter* sp. J44 formed a separate cluster from other *Aminobacter* species.

### 3.3. Whole Genome Phylogeny of Aminobacter Strains

A total of 17 non-redundant genome sequences representative of all members of the *Aminobacter* genus available in the NCBI database were combined with the *A. anthyllidis* LMG 26462^T^ genome sequence and used for the whole genome-based phylogenetic analysis (Table 1). Genome-wide phylogenomic relationships between *Aminobacter* members showed the presence of two distinct clades, supported by >70% bootstrap values (Figure 2a). The first clade was formed by the majority (14/17) of *Aminobacter* species. Within this clade, *A. lissarensis* strains were the closest neighbors of the newly sequenced *A. anthyllidis* LMG 26462^T^ strain (Figure 2a). *A. ciceronei* DSM 15910^T^, *A. aminovorans* KCTC 2477, *A. aminovorans* DSM 10368, *Aminobacter* sp. MDW-2 and *Aminobacter* sp. SR38 formed a separate subclade (*A. ciceronei* subclade hereafter) (Figure 2a). Similarly, a small inter-species distance was also noticed between *Aminobacter* sp. DSM 101952 and *A. aganoensis* DSM 7051^T^ and between *Aminobacter* sp. MSH1 and *A. niigataensis* DSM 7050^T^ (Figure 2a). *Aminobacter* sp. AP02 did not cluster with any validly published *Aminobacter* species, likely representing a distinct species. Consistent with 16S rRNA gene analysis, *Aminobacter* sp. J15, *Aminobacter* sp. J41 and *Aminobacter* sp. J44 formed a separate clade, unrelated to any *Aminobacter* spp.

### 3.4. ANI and dDDH of Aminobacter Species

ANI and GGDC values of the pairwise comparison between *Aminobacter* strains are shown in Figure 3. Pairwise ANI and GGDC values between *A. ciceronei* DSM 15910^T^ and all members of the *A. ciceronei* subclade, namely *A. aminovorans* KCTC 2477, *A. aminovorans* DSM 10368, *Aminobacter* sp. MDW-2 and *Aminobacter* sp. SR38, was above the species demarcating threshold (>96% for ANI and >70% for GGDC; [53,54,55]), suggesting that all these strains belong to the *A. ciceronei* species (Figure 3). ANI and GGDC values between *Aminobacter* sp. DSM 101952 and *A. aganoensis* DSM 7051^T^, and between *Aminobacter* sp. MSH1 and *A. niigataensis* DSM 7050^T^, were also above the species demarcating threshold (Figure 3). ANI and GGDC values between *Aminobacter* sp. AP02 and all other *Aminobacter* strains are below the species threshold (Figure 3), further supporting the hypothesis that this strain could represent a new *Aminobacter* species. 

Notably, *Aminobacter* sp. J15, *Aminobacter* sp. J41 and *Aminobacter* sp. J44 showed the lowest ANI and GGDC values in any reciprocal comparison with any other *Aminobacter* strains, although the pairwise comparison between them suggested that these three strains belong to the same species (Figure 3). According to 16S phylogeny, *Nitratireductor aestuarii* CGMCC 1.1532^T^ was the closest relative to the three strains (Figure 1). Pairwise ANI and GGDC comparisons between *N. aestuarii* CGMCC 1.1532^T^ and *Aminobacter* sp. J15 (78.5% and 14.7%, respectively), *Aminobacter* sp. J41 (78.5% and 14.7%, respectively) and *Aminobacter* sp. J44 (78.5% and 14.7%, respectively) were below the species identity threshold, suggesting that these strains also represent a new species of genus *Nitratireductor*. 

Based on the above observations, reclassification of some members of the genus is proposed in Appendix A.

### 3.5. Nodulation and Nitrogen-Fixation Genes in Members of the Aminobacter Genus

The N-acyltransferase *nodA* gene, implicated in nodulation, has previously been detected in *A. anthyllidis* LMG 26462^T^ by PCR with *nodA*-specific primers and showed extensive similarity with the *Mesorizobium loti nodA* homolog [11]. By inspecting the *nodA* (J1C56_30280) flanking ORFs in *A. anthyllidis* LMG 26462^T^, additional genes of the *nod* cluster were also identified (see locus tags in Figure 4).

To further characterize the composition and physical arrangement of the *nod* gene cluster and detect *nif-fix* genes in the *A. anthyllidis* LMG 26462^T^ genome, a comparative analysis with *Mesorhizobium japonicum* MAFF 303099^T^ and *M. japonicum* R7A (formerly *Mesorhizobium loti*; [56]) genomes, was performed. The well-characterized *nod* and *nif-fix* gene products from *M. japonicum* MAFF 303099^T^ [57,58] were used as query sequences. All 19 *nif-fix* genes and 24 out of 30 *nod* genes formerly described in *M. japonicum* MAFF 303099^T^ [57], were identified in both *M. japonicum* R7A and *A. anthyllidis* LMG 26462^T^ (Figure 4).

Although several species of rhizobia carry nodulation and nitrogen fixation genes on large plasmids [59], *M. japonicum* R7A and *M. japonicum* MAFF 303099^T^ harbor most of *nod* and *nif-fix* genes in a 501-kb chromosomal symbiosis island [57,58]. IslandViewer prediction revealed the presence of a 500-kb symbiosis island also in *A. anthyllidis* LMG 26462^T^, containing the same *nif-fix* and *nod* genes as those detected in *M. Japonicum* R7A (Figure 4 and Figure 5a). Moreover, a nearly identical arrangement of *nif-fix* and *nod* genes in *A. anthyllidis* LMG 26462^T^ and *M. Japonicum* R7A was observed (Figure 5b; [58]).

The *A. anthyllidis* LMG 26462^T^
*nodABC* genes encode key enzymes for the biosynthesis of the Nod factor backbone. Nod factors are oligosaccharide signal molecules playing a key role in the early stages of nodulation [60]. While the chemical structure of the *A. anthyllidis* LMG 26462^T^ Nod factor is unknown, a number of nodulation genes were detected (*nod*, *nol* and *noe* genes) likely implicated in modification of the Nod factor backbone to generate species-specific signal molecule(s) (Figure 5b). The *nodD1* and *nodD2* genes are predicted to encode regulators of the expression of the *nod* structural genes upon interaction with a flavonoid inducer of Nod factor synthesis, secreted by the symbiotic plant [58,61], whereas *nodIJ* genes encode for a Nod factor secretion system [62]. The symbiosis island of *A. anthyllidis* LMG 26462^T^ also contains *nif*-*fix* genes, including *nifA* which encodes for the nitrogen fixation transcriptional regulator, and *nifKDH* genes involved in the synthesis of the MoFe-nitrogenase complex [58].

Remarkably, nodulation and nitrogen-fixation genes were absent from the genomes of *Aminobacter* species other than *A. anthyllidis* LMG 26462^T^ (Figure 2b).

### 3.6. Methylotrophy Genes in Members of the Aminobacter Genus

Degradation of C1 methylated amines takes place via the alternative methylamine dehydrogenase (MADH) or N-methylglutamate (NMG) pathways [14]. The methylamine dehydrogenase *madH* gene, considered a hallmark of the MADH pathway, was not detected in any *Aminobacter* species (Table 1). Hence, oxidation of methylamines was assumed to be carried out by the NMG pathway. A comprehensive description of genes involved in the NMG pathway, in association with the serine cycle, is available for *Paracoccus aminovorans* JCM 7685^T^, in which all these genes map in a 40-kb methylotrophy island (MEI) located on plasmid pAMV1 [63]. The first step of the methylamine-oxidation pathway relies on seven genes involved in the demethylation of the methylamine precursors (Figure 6a). The second step involves eight genes of the NMG pathway for methylamine oxidation to CH_2_=THF (Figure 6b). The last step includes seven genes involved in the oxidation of CH_2_=THF to CO_2_ (Figure 6c). The C1 intermediates generated by the methylamine-oxidation and the NGM pathways are funneled into the serine cycle to generate acetyl-CoA (Figure 6d).

Starting from the *P. aminovorans* JCM 7685^T^ genome annotation, the presence of genes involved in the methylamine-oxidizing pathway genes was investigated in the *A. aminovorans* DSM 7048^T^ type species. A total of 33 genes implicated in all steps of both the methylamine-oxidation and the serine pathways were detected in the *A. aminovorans* DSM 7048^T^ type species (Figure 7a). Notably, the whole set of both NMG and serine cycle genes was observed for 13 strains referable to all validly published *Aminobacter* species (Figure 7a; Appendix A), whereas they were absent in *Aminobacter* sp. J15, J41 and J44, consistent with their belonging to a different genus (Figure 7a and Appendix A). Moreover, the whole serine cycle was not detected in *Aminobacter* sp. DSM 101952, except *mdh* and *eno* genes, along with the citric acid cycle and glycolysis, respectively.

To investigate the physical localization and the organization of genes involved in the methylamine-oxidation and the serine pathways, *A. aminovorans* KCTC 2477 was selected as a reference strain, since it carries all the genes involved in both pathways (33 genes; Figure 7a) and is characterized by a complete well-refined genome, including the circular chromosome and 4 plasmids (Table 1). Genome inspection revealed that 16 out of 33 genes (48.5%) mapped in the *A. aminovorans* KCTC 2477 chromosome. These included seven genes involved in the demethylation of the methylamine precursors, seven genes involved in the CH_2_=THF oxidation pathway, and the *eno* and *mdh* genes of the serine cycle, which are shared with other fundamental pathways (Figure 7b). The remaining 17 genes, namely nine genes from the serine cycle and eight genes involved in the NMG pathway, were mapped in plasmid pAA01 (Figure 7b).

To determine if *Aminobacter* strains can also fix CO_2_ through the Calvin cycle, the presence of the ribulose-1,5-bisphosphatecarboxylase/oxygenase (RuBisCO) genes were investigated. The RuBisCO genes were only identified in *A. lissarensis* DSM 1086, corresponding to locus tags IHE39_RS24920 and IHE39_RS24925, annotated as ribulose bisphosphate carboxylase small and large subunits, respectively (Figure 2b).

### 3.7. Methyl Halide Utilisation Gene Cluster in Members of the Aminobacter Genus

Methyl halide degradation genes were previously described in *A. lissarensis* CC495^T^ (=DSM 17454^T^) [21] and *A. ciceronei* IMB-1^T^ (=DSM 15910^T^) [20]. Genome inspection showed that both strains shared the same organization of the gene cluster and high similarity at the protein level. In addition to previous work [21], *cmuB* and *metF* were identified in *A. ciceronei* DSM 15910^T^ and *A. lissarensis* DSM 17454^T^, respectively (Figure 8).

CmuA is a corrinoid-binding methyltransferase implicated in the transfer of the methyl group from the methyl halides to the corrinoid Co atom, CmuB transfers the methyl group onto tetrahydrofolate (THF), forming methyl THF, which is ultimately oxidized to the key intermediate methylene THF by MetF (Figure 6e). The *paaE* gene product is a putative ferredoxin—NADP reductase likely implicated in the reduction of the inactive Co(I) to the active Co(II) state of the corrinoid cofactor. CmuC and HutI are putative methyltransferase and imidazolonepropionase enzymes, respectively, whose role in the methyl halide degradation pathway is still uncertain.

To investigate the genetic location of the methyl halide degradation gene cluster, putative plasmids were assembled from short-read data with plasmidSPAdes [36]. Three and four large plasmids were predicted in *A. lissarensis* DSM 17454^T^ and *A. ciceronei* DSM 15910^T^, respectively. One of the three plasmids predicted in *A. lissarensis* DSM 17454^T^ (138,736 bp) showed 100% identity with contig 16 (138,704 bp), containing the methyl halide degradation gene cluster and form I CODH genes, together with previously characterized genes involved in plasmid replication and transfer [4]. Likewise, one of the four *A. ciceronei* DSM 15910^T^ plasmids (151,924 bp) showed 100% identity with contig 18 (151,942 bp) encompassing the methyl halide degradation gene cluster. Additional plasmid features were detected in *A. ciceronei* DSM 15910^T^ contig 18, particularly genes encoding for putative replication functions, namely RepA DNA helicase (HNQ95_RS18770), the RepB DNA primase (HNQ95_RS18775), the RepC DNA binding protein (HNQ95_RS18780), together with *traGDCAFBH* (HNQ95_RS18495-HNQ95_RS18525) and *trbIHGFLKJ* (HNQ95_RS18710-HNQ95_RS18740) genes, presumably implicated in plasmid transfer. The physical association between methyl halide degradation and plasmid replication and conjugal transfer genes, together with data from in silico plasmid assemblies, argues for a plasmid location of the methyl halide degradation cluster. Of note, methyl halide degradation genes were not detected in *Aminobacter* strains other than *A. lissarensis* DSM 17454^T^ and *A. ciceronei* DSM 15910^T^ (Figure 2b).

### 3.8. Carbon Monoxide Dehydrogenase Genes in Members of the Aminobacter Genus

Genome inspection revealed the presence of the form II CODH gene cluster in all *Aminobacter* genomes, as opposed to form I CODH, which was detected in only three genomes, namely in *A. lissarensis* DSM 17454^T^, *A. lissarensis* DSM 1086 and *Aminobacter* sp. AP02 genomes (Figure 2b; Appendix A). The form II CODH structural genes invariably showed the typical *coxSLM* organization, and CoxL contained the distinctive AYRGAGR signature [64], whereas form I CODH structural genes were all organized in the *coxMSL* order, followed by *coxDEF* accessory genes, with CoxL containing the distinctive AYRCSFR signature [4,64]. Additional genes of the form I CODH cluster were *coxG* in *A. lissarensis* DSM 17454^T^ and *coxH* and *coxI* in *A. lissarensis* DSM 1086 [4]. In both *A. lissarensis* DSM 17454^T^ and *A. lissarensis* DSM 1086, the structural genes encoding form I CODH were mapped in putative plasmids [4]. To investigate the genomic location of form I CODH genes (i.e., *coxMSL**DEF*) in *Aminobacter* sp. AP02, putative plasmids were assembled from short-read data with plasmidSPAdes [36]. None of the two predicted plasmids of *Aminobacter* sp. AP02 were associated with contig 4 (343,490 bp), containing the *coxMSLDEF* genes, suggesting a chromosomal location of form I CODH genes.

### 3.9. Glyphosate Oxidation Genes in Members of the Aminobacter Genus

Genome inspection revealed that *A. aminovorans* KCTC 2477 carries 15 *phn* genes encoding for enzymes of the glyphosate oxidation pathway, all showing significant similarity at the protein level with homologs from *Sinorhizobium meliloti* 1021 (GCF_000006965.1) (Figure 9a; [26]). The *phn* genes were mapped on the *A. aminovorans* KCTC 2477 chromosome, though their physical organization resembled that observed in the pSymB plasmid of *S. meliloti* 1021 (Figure 9b; [26]). The glyphosate oxidation pathway involves *phnGHIJKLM* gene cluster which embodies the core components of the C-P lyase pathway (Figure 9b). The DUF1045 gene, encoding a member of the two-histidine phosphodiesterase superfamily, has frequently been found next to *phnM* in several bacteria [26]. While the function of DUF1045 is still uncertain, it has been proposed that it could act as a phosphoribosyl cyclic phosphodiesterase implicated in the hydrolysis of cyclic ribose-phosphate, the product of the C-P lyase reaction [26]. Similarly arranged *phn* genes, including DUF1045, were detected in all *Aminobacter* genomes, with exception of *Aminobacter* sp. J15, *Aminobacter* sp. J41 and *Aminobacter* sp. J44 (Figure 2b; Appendix A).

### 3.10. 2,6-dichlorobenzamide and Atrazine Degradation Genes in Members of the Aminobacter Genus

Previous studies on *Aminobacter* sp. MSH1 reported that 2,6-dichlorobenzamide (BAM) degradation genes map on distinct plasmids: pBAM1 carries the BbdA amidase, responsible for the initial transformation of BAM into 2,6-DCBA [65], whereas pBAM2 harbors a large region for the complete degradation of 2,6-DCBA, encompassing two major gene clusters: *bbdR1B1B2B3CDE* and *bbdR2FGHIJ* [30,66] (Appendix A). Genome inspection of all *Aminobacter* members detected BAM degradation genes only in *Aminobacter* sp. MSH1 (Figure 2b).

*Aminobacter* sp. SR38 was described as an atrazine-degrading strain with the *atzA* gene located on plasmid pSR4 and *atzB*, *atzC* and *trzD* genes on plasmid pSR8 (Appendix A; [42]). These genes were unique to *Aminobacter* sp. SR38, since they were undetectable in other *Aminobacter* genomes (Figure 2b).

## 4. Discussion

Currently, the genus *Aminobacter* comprises six validly published species, namely *A. aganoensis* [2], *A. aminovorans* [2,5], *A. anthyllidis* [11], *A. ciceronei* [10], *A. lissarensis* [10] and *A. niigataensis* [2]. Consistent with previous reports [3,10], the phylogenetic analysis based on 16S rRNA gene sequences clearly delineates the genus *Aminobacter* as a clade distinct from all other members of *Phyllobacteriaceae*. However, overall poor resolution and short inter-species distances were observed in the 16S-based phylogeny, prompting us to infer phylogenetic relationships from genome-scale analysis of all members of the genus *Aminobacter*. For this purpose, the whole genome of *A. anthyllidis* LMG 26462^T^ was de novo sequenced, and comprehensive phylogenomic reconstruction of 17 members of the *Aminobacter* genus, including both type strains and the so far unclassified isolates, was inferred. By combining whole genome-based phylogeny with ANI and dDDH analyses, novel taxonomic relationships were reliably established.

Whole genome-based phylogeny revealed that the closest neighbor of *A. anthyllidis* LMG 26462^T^ is *A. lissarensis*. This phylogenetic relationship does not strictly correlate with previous chemotaxonomic data obtained from the Biolog GEN III MicroPlate assay [4], which provided a quite similar metabolic profile for *A. anthyllidis* LMG 26462^T^ and *A. aminovorans* DSM 7048^T^. However, *A. aminovorans* DSM 7048^T^ was the second nearest neighbor of *A. anthyllidis* LMG 26462^T^, and both shared identical morphological traits, such as cell size and the presence of lophotrichous flagella [4], consistent with established phylogenetic relationships.

Whole genome phylogeny, combined with ANI and dDDH analyses, provided compelling evidence that formerly unclassified *Aminobacter* strains can be assigned to definite *Aminobacter* species, with the only exception of *Aminobacter* sp. AP02, which plausibly represents a new, still uncharacterized species. Moreover, the nearly identical *Aminobacter* sp. J15, *Aminobacter* sp. J41 and *Aminobacter* sp. J44 strains were phylogenetically distant from all members of the genus *Aminobacter* and were taxonomically allocated within the genus *Nitratireductor* by 16S rRNA gene analysis. Based on both ANI e dDDH values, *A. aminovorans* DSM 10368, *A. aminovorans* KCTC 2477, *Aminobacter* sp. MDW-2 and *Aminobacter* sp. SR38 could be assigned to the species *A. ciceronei*. In addition, ANI and dDDH results classified *Aminobacter* sp. DSM 101952 and *Aminobacter* sp. MSH1 as *A. aganoensis* and *A. niigataensis*, respectively (Appendix A).

The genus *Aminobacter* comprises a group of environmental bacteria that thrive in polluted soil, with a single species capable of establishing symbiotic interactions with plants. Indeed, *A. anthyllidis* LMG 26462^T^ was isolated from root nodules of *Anthyllis vulneraria* (*Fabaceae*) and showed nitrogen fixation properties [11]. In this study, accurate inspection of the *A. anthyllidis* LMG 26462^T^ genome unraveled the presence of a symbiosis island containing both nodulation and nitrogen-fixation genes, identical to those identified in *M. japonicum* R7A [58]. The advantage of acquiring such a genomic island by *A. anthyllidis* LMG 26462^T^ is huge, as it would allow the exploitation of a new ecological niche, ultimately resulting in “evolution in quantum leaps” [67]. This is because the acquisition of a complete set of *nod-nif-fix* genes, probably by horizontal gene transfer, converted *A. anthyllis* LMG 26462^T^ from a soil saprophyte to a symbiotic nitrogen-fixing bacterium. Profiting from the published *A. anthyllis* LMG 26462^T^ genome sequence, future studies will help to better define the structural and functional organization of the symbiosis island in *A. anthyllidis* LMG 26462^T^.

A metabolic feature common to all *Aminobacter* species is their ability to use certain C1 compounds as the only carbon and energy sources. However, the genetic basis underlying the oxidation and the assimilation of C1 units into biomass have never been investigated in the genus *Aminobacter*. Methylotrophy is the metabolic ability of microorganisms to build biomass and obtain energy from C1 compounds [13,14]. The substrates supporting methylotrophic growth include methane and methanol, as well as methylamines, methyl halides and methylated sulfur species [13]. Members of the *Aminobacter* genus are able to degrade a variety of methylamines and methyl halides, but cannot utilize methane and methanol [1]. The genetic analysis performed in this study provides a comprehensive characterization of the genes involved in methylamine oxidation, starting from its precursors (trimethylamine, trimethylamine-N-oxide, dimethylamine) until complete oxidation to CO_2_. All *Aminobacter* strains, with the exception of *Aminobacter* sp. 15, *Aminobacter* sp. 41 and *Aminobacter* sp. 44, carried genes involved in the methylamine oxidation via the NMG pathway, as well as the serine cycle genes for the assimilation of C1 units. A plasmid location of all genes of the NMG pathway, together with essential genes of the serine cycle, was predicted in *A. aminovorans* KTCT 2477. The co-localization of NMG pathway and serine cycle genes in the same plasmid suggests their assembly as a methylotrophic module that could favor the metabolic efficiency, and thus undergo positive selection [14]. Besides the methylamine pathway, *A. lissarensis* DSM 17454^T^ and *A. ciceronei* DSM 15910^T^ carried genes for methyl halide oxidation to CH_2_=THF, which is then assimilated in the serine cycle or further oxidized to CO_2_.

Since all *Aminobacter* species are equipped with a complete set of serine cycle genes and all but one lack genes encoding for key enzymes of the autotrophic Calvin cycle (i.e., RuBisCO), they can be classified as heterotrophic methylotrophs [68]. The only exception is *Aminobacter* sp. DSM 101952 also lacked essential genes of the serine cycle, and should therefore be considered a non-methylotrophic bacterium, given that the NMG pathway is also present in non-methylotrophs, which employ it for utilization of methylamine as a nitrogen source [69]. Moreover, since *A. lissarensis* DSM 1086 carries both serine and Calvin cycle genes, it can be classified as a facultative autotrophic methylotroph [68]. This bacterium, together with *A. lissarensis* DSM 17454^T^ and *Aminobacter* sp. AP02, also showed carboxydotrophic properties, due to the presence of genes encoding form I CODH. Two CODH forms are known, designated I and II, sharing the same nomenclature (CoxL, CoxM and CoxS subunits) but differing in sequence [22]; form I specifically oxidizes CO with high affinity, whereas form II has a lower affinity for CO and still uncertain function [64], and it was found in all members of the genus *Aminobacter*.

Methylamines are organic nitrogen compounds widespread in the atmosphere. Their deposition may constitute a substantive input of atmospheric nitrogen to terrestrial and aquatic ecosystems [15,70]. A prominent source of methylamine compounds is provided by agricultural systems, where a variety of aliphatic amines, including methylamines, can be emitted in the atmosphere from animal husbandry [71] and biomass burning [72]. Therefore, understanding the sources and sinks of these gases in the environment will contribute to better assess their impact on public health and ecosystem function. Data from this study expand previous knowledge on the methylotrophic metabolism in genus *Aminobacter* and shed more light on the role of methylamine oxidation via the NMG pathway in soil bacteria.

Soil bacteria are highly adaptable organisms, able to survive in extremely difficult conditions. They have evolved smart strategies for surviving during nutritional stress, including the expression of specialized enzyme systems that allow them to grow on rare nutrient sources. Inorganic phosphate (Pi) is a limiting factor in many ecosystems, and phosphonates represent organic compounds containing Pi [25]. Many soil bacteria can degrade phosphonates via the C-P lyase pathway, an enzymatic process responsible for the cleavage of the phosphonate C-P bond, resulting in the formation of N-methylglycine (sarcosine) and a phosphorus-containing molecule [26]. Through this enzymatic pathway, bacteria are also able to degrade the herbicide glyphosate [26,73]. The *phnJ* gene, formerly detected in *A. aminovorans* KCTC 2477 [24], was used by us as a probe to search for the C-P lyase pathway in *Aminobacter* genomes. Our investigation revealed that all *Aminobacter* species, except *Aminobacter* sp. J15, *Aminobacter* sp. J41 and *Aminobacter* sp. J44, carried a complete set of genes for the C-P lyase pathway (*phn* genes) in their chromosome, showing a similar organization as that observed in *S. meliloti* 1021 pSymB plasmid [26]. It can be speculated that the C-P lyase pathway is expressed under conditions of limiting Pi, to enable its acquisition from phosphonates.

Two unclassified *Aminobacter* spp. were previously reported to have a potential role in the bioremediation of additional xenobiotic compounds, namely the atrazine herbicide [28] and the 2,6-dichlorobenzamide (BAM) water pollutant [74]. *Aminobacter* sp. SR38 (*A. ciceronei*, according to this study) degrades atrazine [29] via plasmid-borne *atz* genes [75], whereas *Aminobacter* sp. MSH1 (*A. niigataensis*, according to this study) is the only known strain capable of mineralizing BAM to CO_2_ [41,76], through catabolic enzymes encoded by *bbd* genes carried on a plasmid [30]. The ability to degrade these xenobiotic compounds is unique of these two *Aminobacter* strains, consistent with the location of degradative genes onto extrachromosomal DNA elements.

## 5. Conclusions

This study provides a comprehensive taxonomic reassessment of all species and strains formerly referred to as the genus *Aminobacter*. The de novo sequencing of *A. anthyllidis* LMG 26462^T^ genome combined with an in-depth phylogenomic investigation of *Aminobacter* genus made it possible to (i) refine the evolutionary relationships between six validly published *Aminobacter* species, (ii) assign formerly uncharacterized *Aminobacter* strains to a definite species and (iii) presumptively identify new *Aminobacter* species. Overall, members of *Aminobacter* appear to be metabolically versatile organisms characterized by broad assimilatory and catabolic potentials. These bacteria are endowed with degradative and assimilatory properties which may contribute to the environmental C, N and P cycles. Here, evidence is provided that some metabolic pathways may have been acquired by horizontal gene transfer, involving either genomic island or plasmids, as in the case of genes implicated in nodulation and nitrogen fixation, methylamine and CO oxidation, the serine pathway, methyl halide, 2,6-dichlorobenzamide and atrazine detoxification. Horizontal gene transfer is the main driver of bacterial evolution [77], and it has probably modeled the genome of *Aminobacter* species to broaden their metabolic potential. The adaptive evolution of *Aminobacter* species to face challenging environmental conditions, including fluctuations in nutrients availability or exposure to toxic compounds, holds promise for the employment of these bacteria in bioaugmentation and bioremediation processes.

## Figures and Tables

**Figure 1 microorganisms-09-01332-f001:**
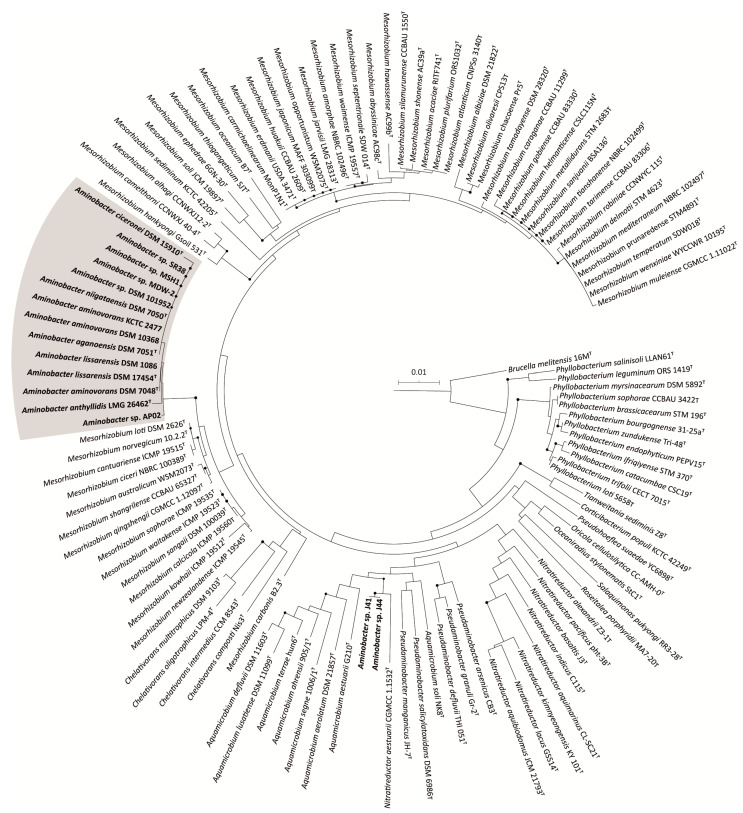
Rooted NJ tree based on the alignment of 1098 bp of 16S rRNA gene sequences, showing the phylogenetic position of the *Aminobacter* genus (highlighted in grey) within the *Phyllobacteriaceae* family. All *Aminobacter* strains are in bold. Bootstrap values ≥ 70% (1000 replicates) are illustrated by black, filled circles at the nodes. *B. melitensis* 16M^T^ was used as an outgroup. The scale bar indicates the expected number of substitutions per site.

**Figure 2 microorganisms-09-01332-f002:**
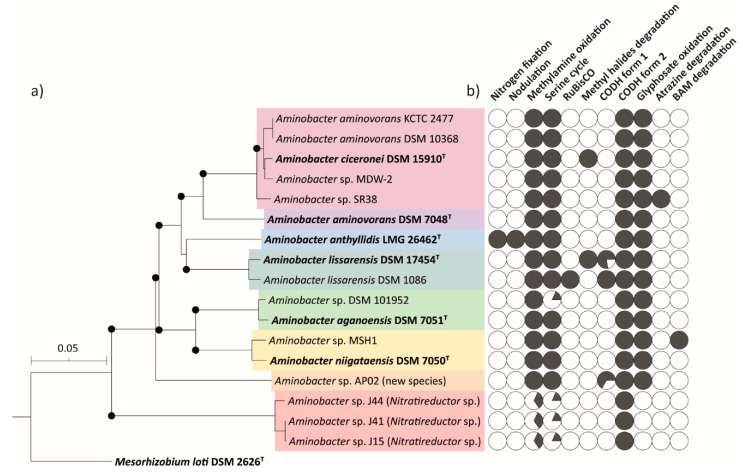
(**a**) Whole-genome phylogeny of the 17 *Aminobacter* strains, including all type strains of validly published species. *M. loti* DSM 2626^T^ was used as an outgroup. The branch lengths are scaled in terms of GBDP distance formula d_5_. Filled circles at the nodes are GBDP pseudo-bootstrap support values > 70% from 100 replications. The scale bar indicates the number of substitutions per variable site. Each colored box contains *Aminobacter* members belonging to the same species, according to ANI and GGDC values (Figure 3); type strains are in bold. (**b**) Black sectored circles denote the presence and degree of completeness of genes involved in the metabolic pathways shown on top of the figure (each gene accounts for 1/22 sector for methylamine oxidation; 1/11 sector for serine cycle; 1/9 sector for CODH form I). White circles denote the complete absence of genes involved in the pathway.

**Figure 3 microorganisms-09-01332-f003:**
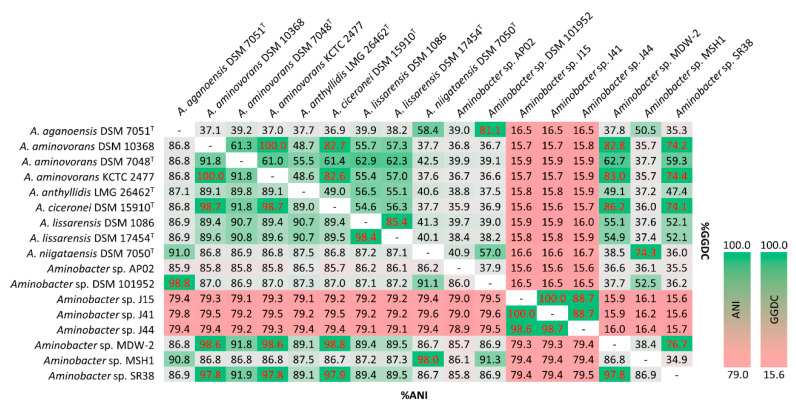
Results of ANI calculation using FastANI and dDDH for the available genomes of *Aminobacter* strains. Text in red indicates ANI and GGDC values exceeding the threshold for species delineation (>96% for ANI and >70% for GGDC).

**Figure 4 microorganisms-09-01332-f004:**
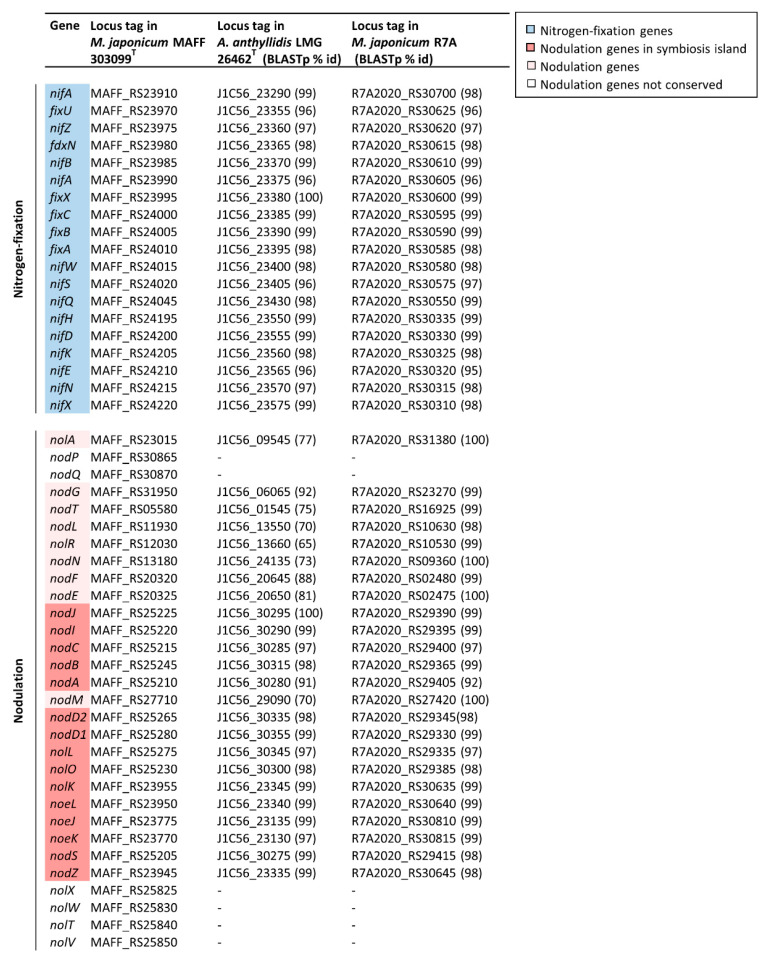
Summary results of BLASTp searches of the translated nucleotide sequences of nitrogen-fixation and nodulation genes from *M. japonicum* MAFF 303099^T^ against *A. anthyllidis* LMG 26462^T^ and *M. japonicum* R7A homologs. Putative genes are colored according to *nod/nif-fix* pathways (see key), where lighter color denotes genes that are not predicted in the symbiosis island, and white denotes the genes not conserved in *A. anthyllidis* LMG 26462^T^ and *M. japonicum* R7A. Hyphen indicates not detected genes.

**Figure 5 microorganisms-09-01332-f005:**
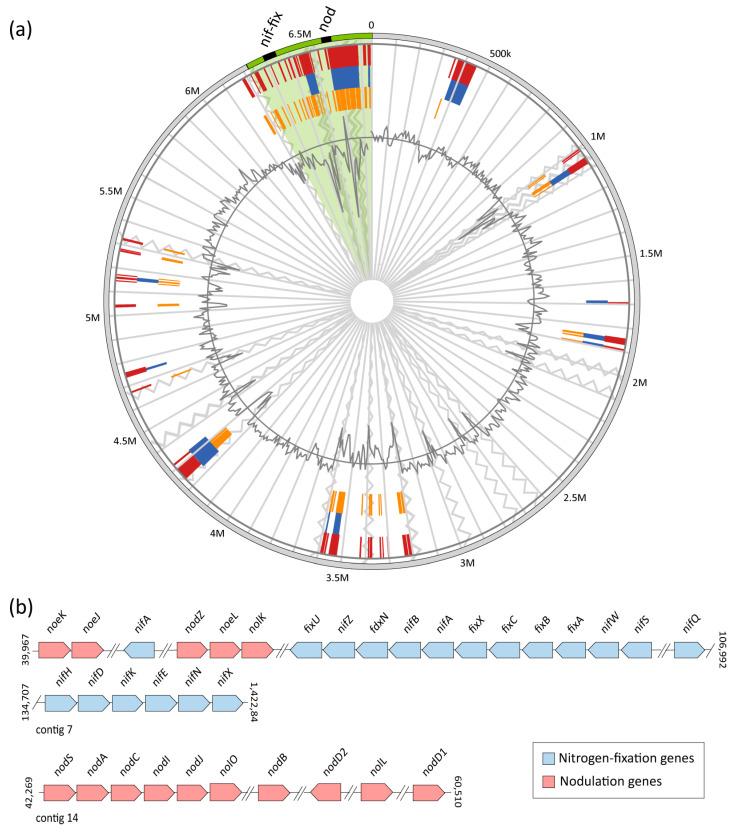
(**a**) Genomic island prediction in the *A. anthyllidis* LMG 26462^T^ genome. The draft genome of *A. anthyllidis* LMG 26462^T^ was aligned on the reference genome of *Aminobacter* sp. SR38. Circular visualization of the predicted genomic island is shown, with blocks colored according to the prediction method; IslandPath-DIMOB (blue) and SIGI-HMM (orange), as well as the integrated results (dark red). The outermost circle (grey) represents aligned contigs, while a region of unaligned contigs, which corresponds to a ca. 500 kb genomic island, is highlighted in green. Black zig-zag line demarcates each of the 30 contigs identified in *A. anthyllidis* LMG 26462^T^. The innermost black plot represents the GC content (%). The *nod* and *nif-fix* gene clusters are shown in black within the symbiosis island region in the outermost circle; (**b**) Organization of nitrogen-fixation and nodulation genes predicted in the *A. anthyllidis* LMG 26462^T^ symbiosis island, colored according to *nod/nif-fix* pathways (see key). Coordinates are numbered according to the first base of each contig. Double slash marks denote genome regions that are not shown. The gene map is not to scale.

**Figure 6 microorganisms-09-01332-f006:**
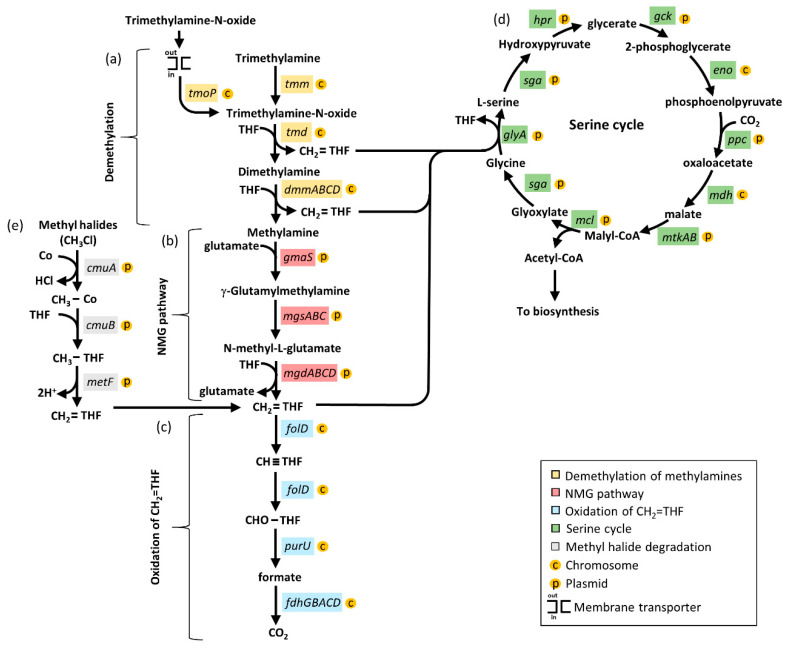
Methylotrophic metabolism in the *Aminobacter* genus. Genes are colored according to the metabolic pathway (see key). (**a**) Demethylation of methylamines: *tmm*, trimethylamine monooxygenase; *tmd*, trimethylamine-N-oxide demethylase; *tmoP*, trimethylamine-N-oxide permease; *dmmDABC*, subunits of the putative dimethylamine monooxygenase. (**b**) NMG pathway: *gmaS* glutamate-methylamine ligase; *mgsABC*, N-methylglutamate synthase subunits; *mgdABCD*, N-methylglutamate dehydrogenase subunits. (**c**) CH_2_=THF oxidation: *folD*, 5,10 methylene tetrahydrofolate dehydrogenase/methenyl tetrahydrofolate cyclohydrolase, *purU* formyl tetrahydrofolate deformylase; *fdhGBACD*, NAD-dependent formate dehydrogenase subunits. (**d**) Serine cycle: *glyA*, serine hydroxymethyltransferase; *sga*, serine glyoxylate aminotransferase; *hpr*, hydroxypyruvate reductase, *gck*, glycerate 2-kinase; *eno*, enolase; *ppc* phosphoenolpyruvate carboxylase; *mdh*, malate dehydrogenase; *mtkAB*, malate-CoA ligase; *mcl*, malyl-CoA lyase. The *scyR* (serine cycle regulator) gene encoding a putative transcriptional regulator is not shown in figure. (**e**) Methyl halide degradation: *cmuA*, methyltransferase I; *cmuB*, methyltransferase II; *metF*, 5,10-methylenetetrahydrofolate reductase. Orange circles indicate the location in the *A. aminovorans* KCTC 2477 genome (C for chromosome and P for plasmid).

**Figure 7 microorganisms-09-01332-f007:**
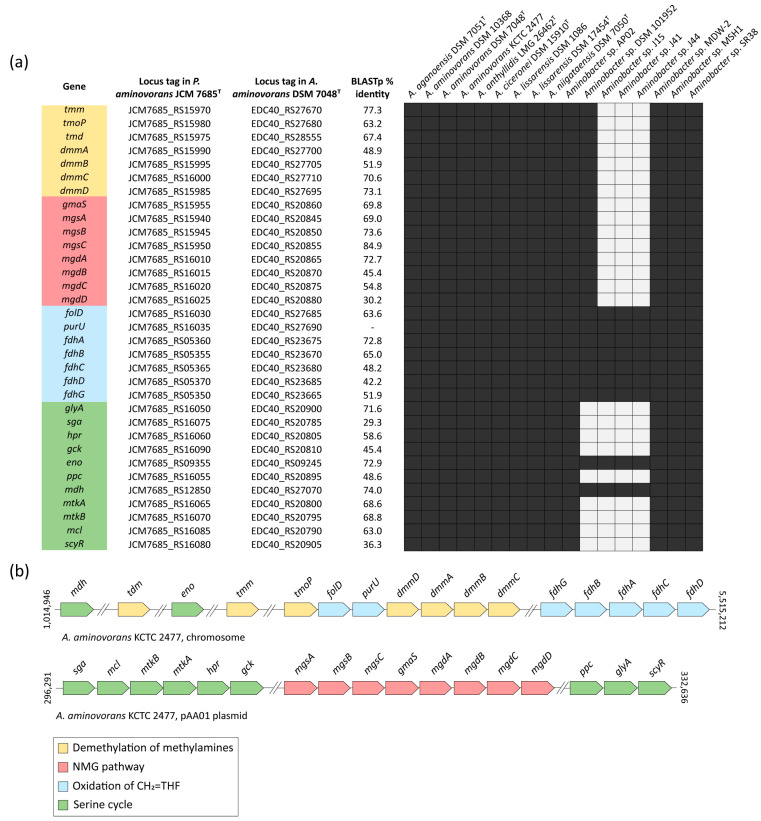
(**a**) Summary results of BLASTp searches of translated nucleotide sequences from the methylotrophy gene clusters from *P. aminovorans* JCM 7685^T^ in *A. aminovorans* DSM 7048^T^. Presence/absence of each gene in all *Aminobacter* genomes is represented by black/white blocks. Putative genes are colored according to the metabolic pathways (see key); (**b**) Methylamine degradation genes organization from *A. aminovorans* KCTC 2477. Genes are colored according to the metabolic pathways (see key). Double slash marks denote genome regions that are not shown. Coordinates are numbered according to the first base of each contig. The map is not to scale.

**Figure 8 microorganisms-09-01332-f008:**
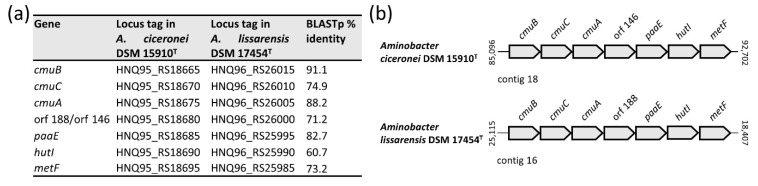
(**a**) Summary results of BLASTp comparison of translated nucleotide sequences from methyl halide degradation genes from *A. lissarensis* DSM 17454^T^ and *A. ciceronei* DSM 15910^T^; (**b**) Physical organization of methyl halide degradation genes in *A. lissarensis* DSM 17454^T^ and *A. ciceronei* DSM 15910^T^; *cmuA*, methyltransferase I; *cmuB*, methyltransferase II; *metF*, 5,10-methylenetetrahydrofolate reductase; *cmuC*, *paaE* and *hutI* are genes of uncertain function. Coordinates are numbered according to the first base of each contig. The map is not to scale.

**Figure 9 microorganisms-09-01332-f009:**
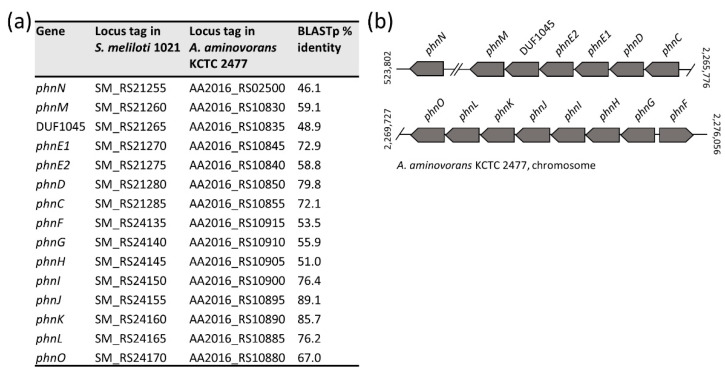
(**a**) Summary results of BLASTp comparison of translated nucleotide sequences from glyphosate oxidation genes from *S. meliloti* 1021 and *A. aminovorans* KCTC 2477; (**b**) Glyphosate oxidation genes organization of *A. aminovorans* KCTC 2477; *phnGHIJKLM* core genes of the C-P lyase enzyme system; *phnCDE1E2*, phosphonate transporter subunits; *phnN*, ribosyl bisphosphate phosphokinase; *phnF*, transcription regulator; *phnO*, aminoalkylphosphonate N-acetyltransferase; DUF1045, putative phosphoribosyl cyclic phosphodiesterase ([26]; see also MetaCyc Pathway: glyphosate degradation III, https://biocyc.org/META/NEW-IMAGE?type=PATHWAY&object=PWY-7807 last accessed on 14 May 2021). Double slash marks denote regions that are not shown. Coordinates are numbered according to the first base of the genome. The gene map is not to scale.

**Table 1 microorganisms-09-01332-t001:** Genome sequence features of selected *Aminobacter* strains.

Organism Name	Genome Size (bp)	Genome Coverage (n×)	No. of Contigs	N50 (bp)	G + C (%)	ORFs (no.)	tRNA Genes (no.)	rRNA Genes (no.)	Source/Country	Accession No.	Ref.
*A. aganoensis* DSM 7051 ^T^	5,765,889	260	52	248,423	63.89	5586	46	3	Soil/JP	GCF_014206975.1	[2]
*A. aminovorans* DSM 10368	6,811,076	220	67	190,669	63.14	6605	48	3	Soil/US	GCF_014195595.1	[37]
*A. aminovorans* DSM 7048 ^T^	5,848,363	256	29	467,483	63.19	5658	47	3	Soil/-	GCF_004341645.1	[2,5]
*A. aminovorans* KCTC 2477	6,890,726	144	1 + 4 ^a^		63.14	6613	54	3	Soil/US	GCF_001605015.1	[38]
*A. anthyllidis* LMG 26462 ^T^	6,717,907	113	30	670,596	62.58	6486	51	3	Root nodule/FR	GCF_018555685.1	[11]
*A. ciceronei* DSM 15910 ^T^	6,774,758	214	96	156,335	63.07	6611	51	3	Soil/US	GCF_014138625.1	[10]
*A. lissarensis* DSM 1086	6,291,275	65	31	458,931	62.96	6023	49	3	Soil/RU	GCF_014863355.1	[3,4,39]
*A. lissarensis* DSM 17454 ^T^	6,541,127	229	45	317,634	62.69	6190	46	3	Soil/IE	GCF_014207495.1	[10]
*A. niigataensis* DSM 7050 ^T^	5,287,613	284	26	468,746	63.4	5083	48	3	Soil/JP	GCF_014200015.1	[2]
*Aminobacter* sp. AP02	5,603,683	210	72	253,161	62.05	5360	48	5	*Populus* root/US	GCF_003148805.1	-
*Aminobacter* sp. DSM 101952	5,233,617	286	36	512,353	63.81	5047	48	3	-	GCF_014201895.1	-
*Aminobacter* sp. J15	4,216,557	293	122	94,548	63.32	3992	49	3	-	GCF_007829635.1	-
*Aminobacter* sp. J41	4,234,633	-	113	102,055	63.34	4011	52	3	-	GCF_000526635.1	-
*Aminobacter* sp. J44	4,186,064	250	90	92,806	63.4	3962	51	5	-	GCF_007829415.1	-
*Aminobacter* sp. MDW-2	6,607,828	200	1 + 3 ^a^		63.19	6400	52	6	Soil/CN	GCF_014250155.1	[40]
*Aminobacter* sp. MSH1	6,321,606	900	1 + 7 ^a^		62.89	6180	52	6	Soil/DK	GCF_003063555.1	[41]
*Aminobacter* sp. SR38	7,367,353	290	1 + 8 ^a^		62.89	7146	53	8	Soil/FR	GCF_014843375.1	[42]

^T^ Type strain. ^a^ Chromosome + plasmids number. -, unknown. Abbreviation: CN, China; DK, Denmark; FR, France; JP, Japan; IE, Ireland; RU, Russia; US, United States.

## Data Availability

The whole-genome shotgun project of *A. anthyllidis* LMG 26462^T^ has been deposited at DDBJ/ENA/GenBank under accession number JAFLWW000000000. The version described in this paper is JAFLWW010000000. The raw sequencing reads are available at the Sequence Read Archive under accession number SRR13863412 and are associated with BioProject accession number PRJNA706925. BioSample SAMN18148366.

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
