# Peer review of "Phylogenomic Reconstruction and Metabolic Potential of the Genus Aminobacter"

_microorganisms, 2021, doi:10.3390/microorganisms9061332_

Round 1
Reviewer 1 Report
L23: Provide the quality control done on extracted genomic DNA.
L20 – L147: Provide the software versions of all programs used in all sequence analyses pipelines.
L164: Provide T, Type strain
L536: Correct the species epithet for M. japonicum R7A
Author Response
Response to Reviewer 1 Comments
Point 1: Line 123: Provide the quality control done on extracted genomic DNA.
Response 1: The Authors wish to thank the Reviewer for her/his favourable evaluation and constructive criticism to the manuscript. With regards to the quality controls, main text has been revised to include this sentence on lines 123-127: “The quantity and quality of the extracted DNA were tested using a Thermo Scientific™ NanoDrop 2000 spectrophotometer (NanoDrop Technologies, Thermo Scientific) and by agarose gel electrophoresis, respectively. Additionally, the genomic DNA quality was evaluated by using Agilent TapeStation Systems 2200 (Agilent Technolo-gies, Santa Clara, US).”
Point 2: Line 120 – 147: Provide the software versions of all programs used in all sequence analyses pipelines.
Response 2: We thank the Reviewer for her/his comment. The main text was revised to include the software version of all programs (lines 138, 141, 151, 161, 162, 167, 181, 185, 192, 200, 213).
Point 3: Line 164: Provide T, Type strain
Response 3: According to Reviewer’s suggestion, “T Type strain” was added in line 154 (former line 164).
Point 4: Line 536: Correct the species epithet for M. japonicum R7A
Response 4: We thank the Reviewer for the correction. The species epithet was corrected to M. japonicum R7A (line 532).
Reviewer 2 Report
The manuscript entitled Phylogenomic reconstruction and metabolic potential of the genus Aminobacter is interesting and describing the potential of Aminobacter in metabolism. However, authors needs to revise the manuscript as per the following suggestions.
- Abstract is not giving appropriate information
- The information in introduction is fragmented please revise.
- Methodology are tough to undertsand for the researchers who are in learning stage, it can be improved.
- Phylogenetic tree proper details are lacking in results and discussion.
- Discussion of the paper is not as expected
- please write the conclusion in a separate section
Author Response
Point 1: Abstract is not giving appropriate information.
Response 1: In our opinion the abstract incorporates all essential information. We kindly request the Reviewer to explain which is the missing or the inappropriate information.
Point 2: The information in introduction is fragmented please revise.
Response 2: The genus Aminobacter includes species and strains which have been characterised based on unique features, as well as on common traits, which are all outlined in the introduction. Individual features of the different Aminobacter species have been illustrated in the introduction to set the basis for investigating whether these features are truly unique or are shared by more than one species. May be the Reviewer refers to such analytical description when she/he states that “The information in introduction is fragmented”. We believe that an analytical description of individual members of the genus is fully instrumental to the scientific content of the experimental part of our part.
Point 3: Methodology are tough to understand for the researchers who are in learning stage, it can be improved.
Response 3: The methodology has been described according to conventional standards accepted by scientific community working in the field of molecular taxonomy, phylogenomics and metabolomics. As recommended by Reviewer #1, the software versions of all programs used in sequence analyses was added, to facilitate reproduction of our findings.
Point 4: Phylogenetic tree proper details are lacking in results and discussion.
Response 4: Proper details of the phylogenetic trees are reported on lines 156-181 (Methods), 222-248 (Results), in the caption of figures 1 (lines 262-265) and 2 (276-283) and discussed on lines 497-514.
Point 5: Discussion of the paper is not as expected.
Response 5: We kindly request the Reviewer to outline what is “not as expected” in the discussion so to provide more constructive criticism and specific recommendations.
Point 6: Please write the conclusion in a separate section
Response 6: According to Reviewer’s suggestion, the Conclusion was reported in a separate section (line 610-628).
Round 2
Reviewer 2 Report
can be accepted